# Integrity of the Left Arcuate Fasciculus Segments Significantly Affects Language Performance in Individuals with Acute/Subacute Post-Stroke Aphasia: A Cross-Sectional Diffusion Tensor Imaging Study

**DOI:** 10.3390/brainsci12070907

**Published:** 2022-07-12

**Authors:** Qiwei Yu, Yan Sun, Xiaoyu Liao, Wenjun Qian, Tianfen Ye

**Affiliations:** 1Department of Rehabilitation Medicine, The Affiliated Suzhou Hospital of Nanjing Medical University, Suzhou 215008, China; liaoxy0617@163.com (X.L.); qwenjun005@sina.com (W.Q.); ytf771009@163.com (T.Y.); 2Department of Radiology, The Affiliated Suzhou Hospital of Nanjing Medical University, Suzhou 215008, China; zizixiaoshanshi@sina.com

**Keywords:** stroke, aphasia, the arcuate fasciculus, diffusion tensor imaging

## Abstract

Objective: To investigate the correlation between the left arcuate fasciculus (AF) segments and acute/subacute post-stroke aphasia (PSA). Methods: Twenty-six patients underwent language assessment and MRI scanning. The integrity of the AF based on a three-segment model was evaluated using diffusion tensor imaging. All patients were classified into three groups according to the reconstruction of the left AF: completely reconstructed (group A, 8 cases), non-reconstructed (group B, 6 cases), and partially reconstructed (group C, 12 cases). The correlations and intergroup differences in language performance and diffusion indices were comprehensively estimated. Results: A correlation analyses showed that the lesion load of the language areas and diffusion indices on the left AF posterior and long segments was significantly related to some language subsets, respectively. When controlled lesion load was variable, significant correlations between diffusion indices on the posterior and long segments and comprehension, repetition, naming, and aphasia quotient were retained. Multiple comparison tests revealed intergroup differences in diffusion indices on the left AF posterior and long segments, as well as these language subsets. No significant correlation was found between the anterior segment and language performance. Conclusions: The integrity of the left AF segments, particularly the posterior segment, is crucial for the residual comprehension and repetition abilities in individuals with acute/subacute PSA, and lesion load in cortical language areas is an important factor that should be taken into account when illustrating the contributions of damage to special fiber tracts to language impairments.

## 1. Introduction

Aphasia is a common and devastating consequence of stroke, usually caused by lesions involving the left hemisphere. It was reported that approximately 24–38% of survivors after a stroke suffered from acute aphasia, and the majority developed chronic aphasia, which significantly affected their private and professional life [1,2]. Therefore, the study of post-stroke aphasia (PSA) has been a longstanding prevalent topic in the field of brain neuroscience in recent decades.

When considering the neural mechanisms underlying PSA, most previous studies have mainly focused on the structural integrity and functional activation of cortical areas in language processing [3,4,5]. Progress made by studies using advanced neuroimaging technologies increased our understanding of the mechanisms underlying language impairments following stroke. The current view is that speech and language abilities could be affected by damage to the specific cortical areas and co-occurring damage to the subcortical white matter pathways [6,7].

The arcuate fasciculus (AF), originating from Wernicke’s area and terminating in Broca’s area, is one of the most studied language-related neural tracts [8]. It was proposed that the AF was responsible for the bidirectional information transfer between the temporal and frontal language areas, which implied the importance of the information regarding speech production on language comprehension [9,10]. In 2005, Catani et al. (2005) [11] delineated the two parallel pathways model of the AF, consisting of a classical long segment (LSAF) that directly connected Wernicke’s with Broca’s area and an indirect pathway that encompassed an anterior (ASAF) and a posterior segment (PSAF). The ASAF connected Broca’s with Geschwind’s area (i.e., the inferior parietal lobule), while the PSAF linked Geschwind’s with Wernicke’s area. Functionally, they speculated that the direct pathway involved in phonological processing (e.g., repetition), and the indirect pathway, mediated lexical-semantic mapping (e.g., auditory comprehension and the vocalization of semantic content). Subsequently, Glasser et al. (2008) [12] divided the AF into two segments: one terminated in the posterior superior temporal gyrus (STG) and another projected to the middle temporal gyrus (MTG). They advocated a phonological role (e.g., repetition) of the segment to STG and a lexical-semantic function (e.g., spontaneous speech) of the segment to MTG. As well, Friederici and colleagues (2013) [13] outlined two distinct functional branches within the AF: one connected the posterior STG (Brodmann area, BA 22/Wernicke’s area) to Broca’s area (BA 44, pars opercularis) and involved in complex syntactic processing, while another linked STG/MTG to the premotor cortex (BA 6) and contributed to speech repetition. Although different models of the AF have been proposed, the three-segment model was widely recognized and applied in language research because of the relatively higher detection rates of the AF [8,14,15].

A growing number of studies have attempted to illustrate the linguistic role of the AF and pointed out that the AF might be involved in the natural progress of language processing, and that damage to the AF was associated with various speech and language impairments, such as speech production, verbal fluency, semantic comprehension, repetition, naming, and reading [8,16]. However, most previous studies investigated its linguistic role in the condition that the AF was treated as a uni-functional entity, and no consensus has yet been reached. This may be partially related to the anatomic details of the AF described by Catani et al. (2005) [11] and Glasser et al. (2008) [12]. Thus, subdividing the anatomical structure and clarifying the linguistic role of the AF segments in brain pathologies including stroke is essential, which will advance our understanding of the pathological mechanisms of the disease as well help clinicians when making diagnoses, predicting outcomes, and setting scientific rehabilitative strategies [2,17].

In this study, we did not evaluate the language-related functions of the whole AF tract in aphasia after stroke. We focused on investigating the correlations between damage to the left AF segments based on the three-segment model of Catani et al. (2005) [11] and language performance and aimed to illustrate the linguistic roles of the left AF segments in acute/subacute PSA. Herein, based on the anatomical details and functions proposed previously, we hypothesized that the contributions of different segments of the left AF to language performance in individuals with acute/subacute PSA varied.

## 2. Materials and Methods

### 2.1. Subjects

Twenty-six right-handed patients (20 males and 6 females, mean age = 54.50 years, SD = 11.30, mean duration of education = 10.92 years, SD = 3.19, mean time post-stroke = 39.46 days, SD = 32.02) with various types of aphasia secondary to left hemispheric strokes were recruited in the department of rehabilitation medicine of our hospital between March 2020 and May 2022 (See Table 1). The inclusion criteria were as follows: first-ever left hemispheric stroke with normal consciousness, aged between 18 and 80 years, right-handedness and Chinese speakers, the consistent presence of aphasia (the aphasia quotient (AQ) of western aphasia battery (WAB) < 93.8), and being able to complete a language assessment and MRI scanning. The exclusion criteria included right or bi-hemispheric stroke, history of stroke, concomitant neurological diseases, other aphasic syndromes such as primary progress aphasia, and being unable to tolerate language assessment or MRI examination.

This study protocol was approved by the local institutional review board (IRB), and all participants or their families gave written informed consent.

### 2.2. Language Assessment

All patients received language assessment by a speech-language therapist with the Chinese version of the WAB. The WAB includes both linguistic subtests, including spontaneous speech, auditory comprehension, repetition, and naming, and nonlinguistic subtests, including subtests for reading, writing, praxis, and construction. The linguistic subtests were analyzed in the current study. The AQ of the WAB reflecting the global severity and type of aphasia was calculated according to the following formula: AQ = (spontaneous speech score + auditory comprehension score/20 + repetition score/10 + naming score/10) × 2. An AQ value less than 93.8 was considered to indicate the presence of aphasia, and the aphasia severity was classified into four levels: very severe (0–25), severe (26–50), moderate (51–75), and mild (≥76), according to the AQ value [18].

### 2.3. MRI Acquisition and Preprocessing

MRI data were acquired using a 3-Tesla Siemens Skyra scanner with a standard radiofrequency head-coil. High resolution structural images of the whole brain using a 3-dimensional T1-weighted magnetization-prepared rapid gradient-echo (MPRAGE) protocol (voxel size of 1 × 1 × 1 mm) were acquired according to the following parameters: repetition time = 2300 ms, echo time = 2.98 ms, inverse time = 900 ms, flip angle = 9°, field of view = 256 × 248 mm^2^, slice thickness = 1.1 mm with no inter-slice gap, number of slices = 160, and acquisition time = 5.2 min. The DTI data were acquired using a single-shot echo planar imaging (EPI) sequence in 49 contiguous slices parallel to the anterior–posterior plane according to the following parameters: repetition time = 7200 ms, echo time = 104 ms, flip angle = 90°, field of view = 896 × 896 mm^2^, acquisition matrix = 96 × 96 mm^2^, reconstructed to matrix = 128 × 128 matrix, b = 0, 1000 s/mm^2^, gradient directions = 64, slice thickness/slice spacing = 2.5 mm/2.5 mm, and acquisition time = 8.3 min.

All original data were preprocessed offline using the Oxford Centre for FMRIB Software Library (FSL) 5.0.9 software package (http://fsl.fmrib.ox.ac.uk/fsl/fslwiki/FSL, accessed on 22 March 2022) [19]. First of all, the raw DICOM data files were converted into the NIfTI format, then head motion and eddy currents were corrected with the Eddy_correct tool. After that, skull-stripping and the removal of non-brain tissue were performed with BET v2.1. Finally, the reconstruction of whole-brain diffusion tensors was performed using the diffusion toolbox. Both the processed T1-weighted images and FA maps were spatially normalized into the MNI152 atlas space using FSL’s FLIRT registration tool.

### 2.4. Lesion Overlay Map and Lesion Load

Lesion maps were manually drawn slice-by-slice on the normalized T1 structural images by using MRIcroGL (https://www.mccauslandcenter.sc.edu/mricrogl/, accessed on 1 March 2022), and lesion volumes were calculated by using ITK-SNAP (http://www.itksnap.org/, accessed on 1 March 2022). To evaluate the impact of damage to cortical language areas on language deficits, the lesion load (i.e., the percentage of cortical language areas overlapping with the stroke lesion) was calculated according to the following formula: lesion load = (the volume of the overlap/the total volume of the cortical language areas) × 100% [15,20].

The cortical language areas consist of the frontal language area, including both pars opercularis and pars triangularis (BA 44 and 45); the parietal language area, including the angular gyrus (BA 39); the supramarginal gyrus (both the anterior and the posterior divisions, BA 40); the temporal language area, including the superior temporal gyrus posterior division (BA 22); and the middle temporal gyrus posterior division (BA 21). All these cortical areas were defined and extracted as regions of interest (ROI) based on the Harvard–Oxford Cortical Structural Atlas in the FSL software package [21,22].

### 2.5. Reconstruction of the AF

The diffusion toolkit (version 0.6.4.1) [23] was used for diffusion imaging data reconstruction and fiber tracking, and the TrackVis (version 0.6.1) software was applied to manually delineate the ROIs and perform fiber track visualization on the normalized FA images. A three-ROIs approach was applied to accomplish the virtual dissection of the bilateral AFs based on a deterministic fiber-tracking algorithm (fiber assignment by continuous tracking—FACT algorithm): a frontal ROI (ROI 1, the green 2D disk) was placed on the coronal slice at the entrance to the frontal lobe (anterior to the central sulcus), a parietal ROI (ROI 2, the red 3D sphere) was placed tangent to the inferior parietal cortex, and a temporal ROI (ROI 3, the blue 3D sphere) was placed on the axial slice at the entrance to the temporal lobe (below the Sylvian fissure) [15,24] (see Figure 1). Each segment of the AF was defined by the combination of 2 of these 3 ROIs. The fiber tracts passing through both the ROI 1 and ROI 2 but not ROI 3 were classified as the ASAF, the streamlines passing through both the ROI 2 and ROI 3 but not ROI 1 formed the PSAF, and the streamlines passing through both the ROI 1 and ROI 3 but not ROI 2 constituted the LSAF [14]. Fiber tracking was initiated with a minimum fractional anisotropy (FA) value at 0.20 and an angle threshold of 35°.

To identify the anatomical alignment and spatial location, the reconstructed three-dimensional segments of the bilateral AFs and stroke lesions were visualized on each spatially normalized T1-weighted image (Figure 2). Then, the patients were classified into three groups according to the reconstruction of the left AF [25,26]: group A, completely reconstructed; group B, non-reconstructed; and group C, partially reconstructed. The mean FA value and fiber number of the left ASAF, PSAF, and LSAF were measured for statistical analyses.

### 2.6. Statistical Analyses

Statistical analyses were performed using the IBM SPSS software package (version 25.0). The normality of data was evaluated by the Shapiro-Wilk test. The Pearson correlation and one-way ANOVA were used for parametric variables (normally distributed data with homogeneity of variance), whereas the Spearman correlation analysis and Kruskal–Wallis test with the Nemenyi post-hoc test were used for nonparametric variables (non-normally distributed data or with a heterogeneity of variance). The significance level in multiple comparisons was adjusted with the false discovery rate (FDR) correction. The correlation coefficient indicates the strength (weak correlation, 0.1–0.29; moderate correlation, 0.3–0.49; and strong correlation, ≥0.50) and direction (positive or negative) of the relationship between two variables [27]. The statistical significance level was set at 0.05.

## 3. Results

### 3.1. Demographic and Clinical Characteristics

The demographic and clinical data of all patients arere shown in Table 1. All patients completed the language assessment and MRI sequence. The type of aphasia was classified into non-fluent aphasia (16 cases), including Broca’s aphasia; global aphasia; mixed transcortical aphasia (MTA); transcortical motor aphasia (TMS); and fluent aphasia (10 cases), including Wernicke’s aphasia, conduction aphasia, anomic aphasia, and transcortical sensory aphasia (TSA). According to the criteria [18], the severity grade of the patients’ aphasia ranged from mild to very severe.

### 3.2. Correlation Analyses between Language Performance and MRI Measures

First, we estimated the correlations between demographic and stroke-related variables and language performance. Pearson correlations were used to analyze the correlation between age and AQ, spontaneous speech, comprehension, naming, and fluency, respectively, and Spearman correlations were applied to analyze the correlations between education, time post of stroke, and language performance, because of non-normally distributed variables. For dichotomous variables (sex and type of stroke), the independent sample *t*-test or Mann–Whitney U test was conducted to compare the intergroup difference in language performance. Both demographic and stroke-related variables were not significantly related to any language subset of WAB (*p* > 0.05).

Subsequently, we analyzed the correlations between MRI metrics and language performance using Spearman correlation due to non-normally distributed variables. Lesion volume was significantly negatively related to spontaneous speech, comprehension, and AQ, and lesion load was significantly negatively related to most language subsets other than fluency. The FA values in the left PSAF and LSAF were significantly positively associated with any language subsets other than fluency and/or repetition, and the fiber number of the left posterior and LSAF was significantly positively associated with AQ, comprehension, repetition, and/or spontaneous speech. However, when lesion load was set as a controlled variable, partial correlations revealed that the PSAF was significantly strongly related to comprehension, repetition, naming, and AQ, while the LSAF was significantly moderately related to comprehension, naming, and AQ (Figure 3). No significant correlation was found between the ASAF and language measures. Due to its strong relationship with the lesion load of cortical language areas, the lesion volume variable was not entered into the partial correlation analyses. The results of the correlation analyses are presented in Table 2 and Table 3.

Next, the correlations between demographic and stroke-related variables, MRI metrics, and language performance were further estimated in the conditions of different reconstructions of the left AF.

### 3.3. Intergroup Demographic and Stroke-Related Variables Analyses

In terms of intergroup differences in demographic and stroke-related variables, a significance in the severity of aphasia was found only between group A and group B (*p* < 0.05). No significant difference was found among groups in any other demographic and clinical metrics.

### 3.4. Lesion Overlay Map and Lesion Load Analyses

The lesion volumes of all patients: range = 1.44–200.50 cm^3^, mean = 47.07 ± 49.60. The lesion volumes among groups: group A, range = 7.94–75.34 cm^3^, mean = 34.41 ± 25.83; group B, range = 21.02–200.50 cm^3^, mean = 92.30 ± 68.68; and group C, range = 1.44–151.30 cm^3^, mean = 34.92 ± 39.30. The sum volume of the cortical language areas extracted from the cortical structural atlas within the FSL software package was 51.05 cm^3^. The lesion load of all patients was as follows: range = 0–48.30%, mean = 9.64 ± 13.66. The lesion load among groups was as follows: group A, range = 0–26.88%, mean = 3.83 ± 9.37; group B, range = 1.97–35.17%, mean = 15.03 ± 13.40; and group C, range = 0.03–48.30%, mean = 10.83 ± 15.65.

There were significant differences in the lesion load of the language areas among the three groups of patients (*p* < 0.05). The Kruskal–Wallis and Nemanyi post-hoc tests showed that the lesion load in group A was significantly smaller than in group B (adjusted *p* < 0.05), but there was no significant difference between group A and group C or between group B and group C. No significant difference in lesion volume was found among groups. The lesion overlay map of all patients (see Figure 4a) shows the heterogeneous distribution of stroke lesion locations among the patients. The lesion maps of groups are presented in Figure 4b.

### 3.5. Intergroup Language Performance Analyses

Significant differences were observed among groups in comprehension (F = 7.943, *p* = 0.002), repetition (*p* = 0.004), naming (F = 3.858, *p* = 0.036), and AQ (F = 7.016, *p* = 0.004) but not in spontaneous speech nor fluency subsets. Post-hoc tests demonstrated that patients in group A and group C had higher scores of comprehension and repetition subsets than those in group B and that AQ value and naming scores in group A were higher only than those in group B. No significant differences in language subsets were observed between group A and group C. The results of language performance analyses were shown in Table 4 and Table 5 and Figure 5.

### 3.6. Intergroup Diffusion Indices Analyses

We attempted to reconstruct the segments of bilateral AFs in all patients according to the approach mentioned. As a result, the left AF was completely reconstructed in 8 cases (group A, patient 01–08), non-reconstructed in 6 cases (group B, patient 09–14), and partially reconstructed in 12 cases (group C, patients 15–26). In group C, only the ASAF was non-reconstructed in two cases (patients 15–16), only the PSAF was non-reconstructed in two cases (patient 17–18), and only the LSAF was non-reconstructed in two cases (patient 19–20); however, both the ASAF and LSAF were non-reconstructed in four cases (patient 21–24), and both the PSAF and LSAF were non-reconstructed in two cases (patient 25–26). In total, the ASAF was non-reconstructed in six cases (50%), the PSAF was non-reconstructed in four cases (33%), and the LSAF was non-reconstructed in eight cases (66.7%). In addition, despite being reconstructed, different degrees of damage to the residual portion of the left AF segments were observed in the patients of group C. The reconstruction of the AF segments was presented in Figure 2.

Because of the non-normal distribution of the DTI parameters in the left AF (*p* < 0.05), the independent samples Kruskal–Wallis test was applied to estimate the significance among groups. The results showed significant intergroup differences in both the FA value and fiber number in all segments of the left AF. Post-hoc tests revealed that the diffusion indices on the anterior and LSAF in group A were significantly higher than those in group B and group C and that the diffusion indices on the PSAF in group A and group C were significantly higher than those in group B. Neither significance in the PSAF was found between group A and group C, nor in the anterior and LSAF between group B and group C. The results of intergroup diffusion indices analyses are shown in Table 6 and Table 7.

## 4. Discussion

In the current study, to investigate the linguistic role of the left AF segments based on a three-segment model in aphasia, we assessed the correlations between demographic and stroke-related variables and language performance and further analyzed the intergroup differences in both language performance and diffusion indices in the condition of different reconstructions of the left AF segments in patients with acute/subacute aphasia secondary to left hemispheric stroke. Our results could be translated into two findings: (i) the integrity of the left AF segments, particularly the PSAF, is crucial for the residual auditory comprehension and repetition abilities in individuals with acute/subacute PSA; and (ii) the lesion load of cortical language areas may be an important factor that should be taken into account when illustrating the contributions of damage to special fiber tracts to language impairments. Below, we discussed the linguistic functions of the left AF segments and our findings, respectively.

### 4.1. The Importance of Lesion Load of Cortical Language Areas in PSA

Correlation analyses revealed a significant linguistic role of the lesion load of the cortical language areas in aphasia; we hence first discussed the contribution of lesion load to language deficits.

Lesion load is defined as a combined variable of the lesion site, and lesion size is an important parameter in brain injury research and is commonly used to measure the effects of a lesion on anatomical structures [28,29,30]. It has been reported that the relationship between the disruption of special fiber tracts and language impairments can be mediated by damage to cortical areas. Breier et al. (2008) [31] found that when considering the impacts of lesions extended into the left temporal lobe, the significant relationship between damage to the left AF and comprehension deficits did not exist, namely, the relationship between the disruption of the AF and comprehension deficits was not independent of the damage to the cortical language areas. Similarly, Ivanova et al. (2021) [15] recently pointed out that the relationship between fiber tract indices and language performance significantly diminished when lesion size was considered. Consistent with these previous findings, our results also highlighted the important influence of damage to cortical language areas in the correlation between the left AF segments and language performance after a stroke. Therefore, the lesion load effects of cortical areas should be taken into account to discriminate between the differential contributions of special fiber tracts and the observed speech disturbances, when illustrating the functions of fiber tracts [15]. However, the extent to which language impairments were directly an outcome of damage to the cortical areas involved remained unclear. We also did not assess the effects of the lesion site on language impairment as they were reported to be other important factors in PSA [28,32]. Additionally, several studies reported that the lesion load of the left AF might be used as a variable to predict impairment of speech production, speech fluency, and naming abilities in individuals with PSA [33,34]. However, it was proposed that disconnection of the tract was generally more sensitive than the lesion load of the tract when evaluating white matter damage [35,36].

When analyzing the intergroup differences in language performance and lesion load, it seemed that these intergroup differences in language performance could not be sufficiently explained by the impacts of damage to the language areas. For example, patients in group A and group C performed better in language tests than those in group B, while the lesion load in group C was not significantly different from that in group B. Although lesion size was reported to be significantly related to behavioral impairments [37,38], Geva et al. (2015) [39] found that patients whose left AF could be tracked on DTI tractography performed better in language task tests than those the AF could not track, and this difference was significant over and above the influence of lesion size. Accordingly, these intergroup differences in language performance may be somewhat related to the disruption of the left AF.

### 4.2. The Linguistic Roles of the Left ASAF in PSA

In the seminal work of Catani and colleagues (2005) [11], the left ASAF was proposed to contribute to speech production. Since the three-segment model of the AF was proposed, several publications have reported the involvement of the left ASAF injury in speech production and verbal fluency. Fridriksson et al. (2013) [40] argued that damage to the left ASAF negatively influenced speech fluency, which implied a robust predictive role of this subsection in speech fluency impairments in patients with chronic non-fluent aphasia. Basilakos et al. (2014) [41] found that the overlapping portions of the aslant and the ASAF were a significant predictor of fluency in PSA. Van Geemen et al. (2014) [42] reported that the electrostimulation of the left ASAF caused speech production disturbances in patients with a glioma involving the left ventral premotor cortex (vPMC). Recently, Gajardo-Vidal and colleagues (2021) [43] pointed out that both the left ASAF and LSAF were likely to contribute to long-lasting speech production impairments after Broca’s area was damaged.

However, Ivanova et al. (2016) [44] reported no significant correlation between the ASAF and any language scores in patients with chronic PSA. Forkel et al. (2020) [14] found that the left ASAF was not associated with repetition and naming performance in patients with primary progress aphasia (PPA). Although PPA and PSA significantly differ in terms of their underlying etiology and clinical manifestations, this finding does not support the role of the left ASAF in repetition and naming abilities. Similarly, we did not find a significant relationship between the ASAF and language measures, despite a significant intergroup difference in diffusion indices on the ASAF. There are two possible explanations for these contradictory results about fluency: firstly, verbal fluency is a multidimensional parameter of speech production that encompasses various elements, including speech rate, prosody, phrase length, syntactic structure, pauses, articulatory struggle, and accuracy; it is difficult to assess and lacks standard measuring instruments [45,46]. Secondly, we used the 10-point rating scale within the WAB to evaluate verbal fluency, which might insufficiently reflect the multidimensional nature of verbal fluency [40]. We did not determine the relationship between the ASAF and speech production impairments according to the limited evidence since it was reported that a lesion commonly disrupted both the ASAF and the LSAF due to the extreme proximity running in the frontoparietal white matter, which possibly resulted in the functional submersion of the ASAF [43,47]. This co-occurring damage to the ASAF and the LSAF can be seen in patients in group C (see Figure 2). Future research should comprehensively assess verbal fluency and try to dissect the roles of the ASAF from the LSAF in language impairments using specialized methods.

### 4.3. The Linguistic Contributions of the Left LSAF in PSA

The LSAF directly connecting the Wernicke’s territory with the Broca’s territory was postulated to subserve the dorsal phonological stream, which was involved in mapping acoustic features into articulatory representations [11,12]. It was reported that the LSAF might be involved in pragmatic integration and higher cognitive function processes of language [48,49]. López-Barroso et al. (2013) [50] pointed out that the direct connections between temporal and frontal areas through the left LSAF contributed to novel word learning by mediating fast communication interactions between auditory and motor areas (i.e., auditory–motor integration). Gullick et al. (2015) [51] found that the left LSAF was the best significant predictor of reading ability change in children between the ages of 8 and 14.

In studies on brain pathologies, Forkel et al. (2020) [14] found no significant correlations between the left LSAF and naming, word comprehension, and repetition deficits in patients with PPA. In the study by Ivanova et al. (2021) [15], the microstructural integrity of the left LSAF was associated with auditory comprehension and naming scores, despite not surviving the FDR correction for multiple comparisons. In this study, although correlation analysis demonstrated a relationship between the LSAF and language performance, including AQ, comprehension, and naming, the specific role of the LSAF could not be further identified in intergroup differences analyses because the diffusion indices on the LSAF in group A were higher than those in group C, while the language performance of patients in group A was not statistically different from those in group C. According to the models of Glasser et al. (2008) [12] and Friederici et al. (2013) [13], the LSAF could be further subdivided into two different branches, respectively, terminating in the posterior STG (pSTG) and the posterior MTG (pMTG). It has been shown that the pSTG activates for sentence-level semantics and the pMTG supports lexical-semantic processes [52,53]. In a recent neuroanatomical framework for syntax, Matchin et al. (2019) [54] proposed that the pMTG, the pSTG, and the posterior inferior frontal gyrus (pIFG) might be involved in the syntactic processing. They pointed out that the pMTG and the pSTG were crucial for both sentence production and comprehension. Accordingly, the LSAF directly connecting the pIFG and the pSTG/pMTG may support the processing of syntactically complex sentences [55]. Indeed, damage to the left LSAF could cause deficits in processing complex syntactic structures, which results in impairments in comprehension of noncanonical sentences [56,57]. Collectively, these prior findings may provide support for the correlations between the left LSAF and comprehension as well as naming performance, which were found in the partial correlation analyses. Thus, the contributions of the LSAF to higher-order language abilities should be further investigated in the future.

According to the classic models of language organization, the left LSAF was associated with speech repetition and its disruption would lead to conduction aphasia [47]. However, a few reports argued that lesions to the left AF were insufficient to cause repetition deficits or unnecessary for recovery from aphasia [58,59]. Obviously, the reports of Forkel et al. (2020) [14] and Ivanova et al. (2021) [15], together with our findings, do not support the role of the left LSAF in repetition. According to the model proposed by Friederici et al. (2012) [55], the LSAF was responsible for the processing of syntactically complex sentences but not speech repetition. This viewpoint also supports our findings. In addition, the high injury rate and heterogeneity of the LSAF (as seen in Figure 2) in patients in group C made it difficult to determine the relationship between the left LSAF injury and specific language subsets. Indeed, it seemed impossible to isolate the linguistic role of the left LSAF in PSA because of common co-occurring damage to the AF segments [43]. Therefore, the contributions of the left LSAF to aphasia after stroke, particularly to speech repetition ability, should be further explored.

### 4.4. The Linguistic Functions of the Left PSAF in PSA

We found a significantly strong relationship between the left PSAF and the residual comprehension and repetition abilities in the current study. The PSAF connects the Geschwind’s with Wernicke’s territory (the center of auditory comprehension). Catani et al. (2005) [11] hypothesized that the indirect segments of the left AF connecting temporal and parietal areas might support auditory comprehension. Lesion-symptom mapping studies have also demonstrated the importance of the temporal portion of the left AF in sustaining lexical-semantic integration [60]. Breier et al. (2008) [31] found that damage to the AF was related to comprehension deficits, which might be mediated by lesions involving the left temporal lobe language areas. Song et al. (2011) [61] pointed out that a lesion involving Wernicke’s area and the left PSAF would lead to Wernicke-like conduction aphasia. Wernicke’s area has long been thought to be critical for language comprehension, and the combination of damage to this area and neighboring regions, including the underlying fiber tracts, commonly results in a special type of aphasia characterized by auditory comprehension deficits. In the DTI studies by Ivanova and colleagues [15,44], the FA value and volume measurements of the PSAF were significantly related to some lexical-semantic and syntactic language abilities, including auditory comprehension. Based on these previous reports and two recent studies that demonstrated that the pSTG, pMTG, and the posterior superior temporal sulcus (pSTS) were crucial for sentence comprehension and phrase comprehension [54,62], it is not novel to state that damage to the temporal cortical language areas and the left PSAF significantly affected auditory comprehension ability.

The viewpoint that the left AF is involved in speech repetition ability, and that damage to this tract commonly causes conduction aphasia characterized by poor repetition, has long been recognized [10,31]. However, which subsection of the left AF is responsible for speech repetition remains unclear. Speech repetition is a complex ability involving the perception of speech, phonological working memory to hold the perceived information, and some aspects of speech production (i.e., articulatory planning and execution) [13]. The networks for speech perception and conceptual-semantic systems are within the MTG, the STG, and the inferior parietal lobes, suggesting the importance of the temporal and inferior parietal lobes on speech repetition [54]. Recently, Forkel et al. (2020) [14] reported that the atrophy of both the temporo-parietal cortex and the indirect pathways of the left AF was prominent in patients with PPA with severe repetition deficits and that the volume of the left PSAF was highly associated with repetition deficits. Similarly, Ivanova et al. (2021) [15] pointed out that the volume of the PSAF contributed to speech repetition in individuals with PSA. Indeed, studies using quantitative lesion mapping have reported that the cortical damages that most frequently result in repetition impairments are located within the left temporo-parietal region and that the inferior parietal lobe has been indicated to encompass the cognitive module necessary for repetition ability [63,64,65]. Therefore, consistent with these findings, our results support the relationship between damage to the left PSAF and comprehension and repetition deficits in acute/subacute PSA. Given the importance of the temporal and parietal lobes for phrase comprehension and sentence syntactic processing [54,62] and its anatomical connection between these two areas, we inferred that the PSAF might play an important role in higher-order language functions (e.g., syntax), which was expected to be verified in future research.

Notably, we did not find a clear correlation between damage to the PSAF and naming deficits when analyzing the intergroup differences in language performance and diffusion indices, which seemed to be inconsistent with the findings of Ivanova et al. (2021) [15]. Naming is a complex process including early visual processing and recognition, the retrieval and selection of semantic knowledge, lexical retrieval, and the coordination and execution of motor plans for the articulators [66]. Partial correlation analyses showed that both the left PSAF and LSAF were significantly associated with naming performance, suggesting an important role of the temporal lobe in naming processing. However, we did not control the impacts of other segments because of the co-occurring damage to the AF segments in the current study. Hence, it is difficult to elucidate the relationship between the left PSAF and performance on naming tasks, according to our results.

Several limitations to this study should be mentioned. First, the number of patients was somewhat limited, especially considering the high co-occurring injury rate and heterogeneity of the ASAF and LSAF in patients in group C. Second, the method of language assessment applied in this study may be insufficient to fully reflect the real language ability of our patients, particularly in terms of verbal fluency, naming, and complex syntax. Third, we did not subdivide the terminal branches of the left LSAF, which resulted in obstacles in investigating its roles in higher-order language functions. Finally, it is a disadvantage of the deterministic fiber-tracking algorithm in crossed fiber tracking, which may affect the reconstruction of the AF. Therefore, future investigations enlarging sample sizes and employing special language assessment instruments as well as applying advanced neuroimaging techniques such as diffusion kurtosis imaging (DKI), diffusion spectrum imaging (DSI), or high angular diffusion magnetic imaging (HARDI) are warranted to clarify which segments of the AF play a decisive role in different language subsets. In addition, the prognosis of our patients’ language abilities is expected to be observed in a longitudinal investigation in the future.

## 5. Conclusions

Several previous studies have reported the linguistic role of the AF segments in the physiological and pathological brain. In this study, we investigated the contributions of damage to the left AF segments to language impairments in individuals with acute/subacute PSA. We found significant correlations between the left AF segments and language performance; particularly, the PSAF seemed to be crucial for the residual comprehension and repetition abilities. Despite several limitations, our findings support the importance of cortical language areas and highlight the linguistic role of the left AF segments in acute/subacute PSA. We believe that is the present topic is of great clinical importance, which is helpful for diagnosis and prognostic prediction in acute/subacute aphasia, particularly in cases where the patients were unable to cooperate with completing clinical assessments. Therefore, the conduct of DTI research on this topic should be encouraged in the future.

## Figures and Tables

**Figure 1 brainsci-12-00907-f001:**
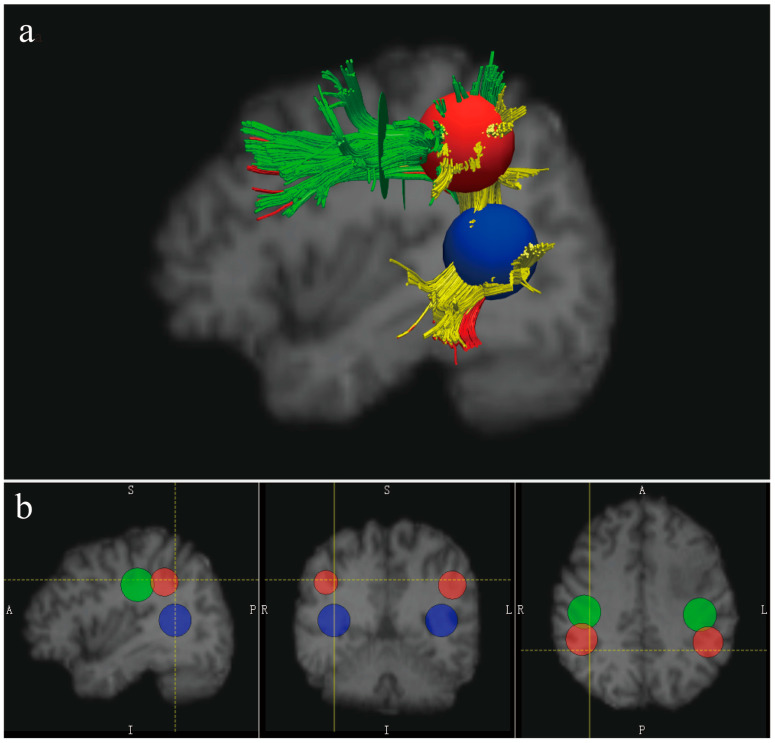
The reconstruction of the bilateral arcuate fasciculus (AF). (**a**) A sample of the reconstructed left AF according to the 3-ROIs approach: AF anterior—green, AF posterior—yellow, and AF long—red; and (**b**) the location of the three ROIs on the normalized T1 images (ROI 1—the green 2D disk, ROI 2—the red 3D sphere, and ROI 3—the blue 3D sphere).

**Figure 2 brainsci-12-00907-f002:**
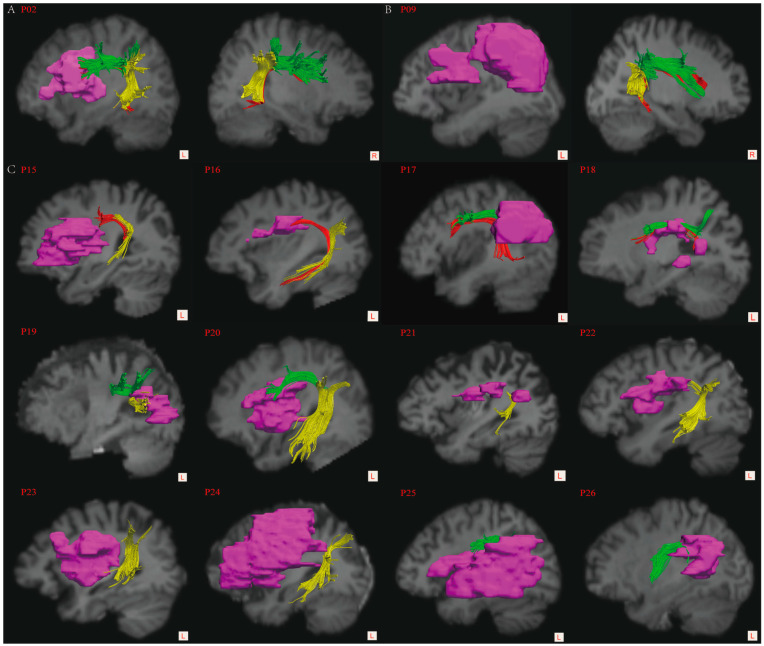
The reconstruction of the left arcuate fasciculus (AF). (**A**) A sample of the completely reconstructed left AF according to the 3-ROIs approach in group A; (**B**) a representative of the non-reconstructed left AF in group B; and (**C**) all patients with the partially reconstructed left AF in group C. The pink blocks—stroke lesion. In group C, the anterior and LSAF presented with different degrees of damage, despite being partially reconstructed, while the PSAF was relatively less damaged, by contrast.

**Figure 3 brainsci-12-00907-f003:**
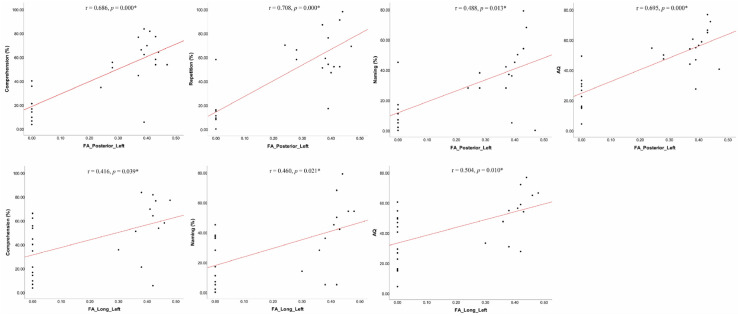
The partial correlation analyses of the PSAF and LSAF and language subsets (controlled variable: lesion load). * Statistical significance level: α = 0.05.

**Figure 4 brainsci-12-00907-f004:**
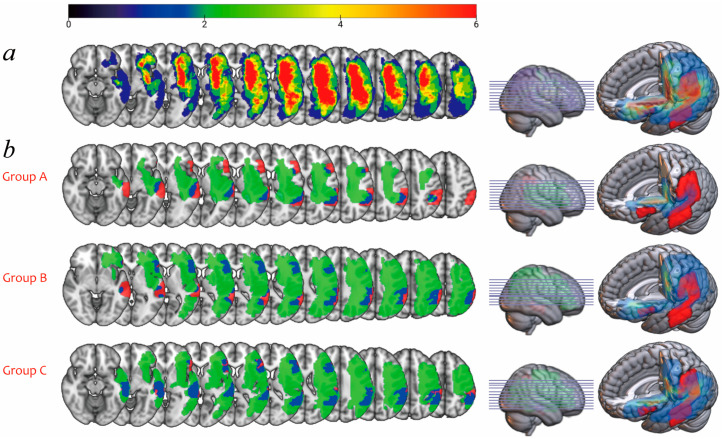
Lesion overlay maps. (**a**) The lesion-overlay map for all 26 patients with PSA. The cold color represents the areas with less lesion overlap, while the hot color represents the areas with high lesion overlap; (**b**) the lesion map of the three groups of patients. The red regions represent the left cortical language areas, the green regions represent the stroke lesion distribution of the patients, and the blue regions represent the overlapping areas between the cortical language areas and stroke lesions.

**Figure 5 brainsci-12-00907-f005:**
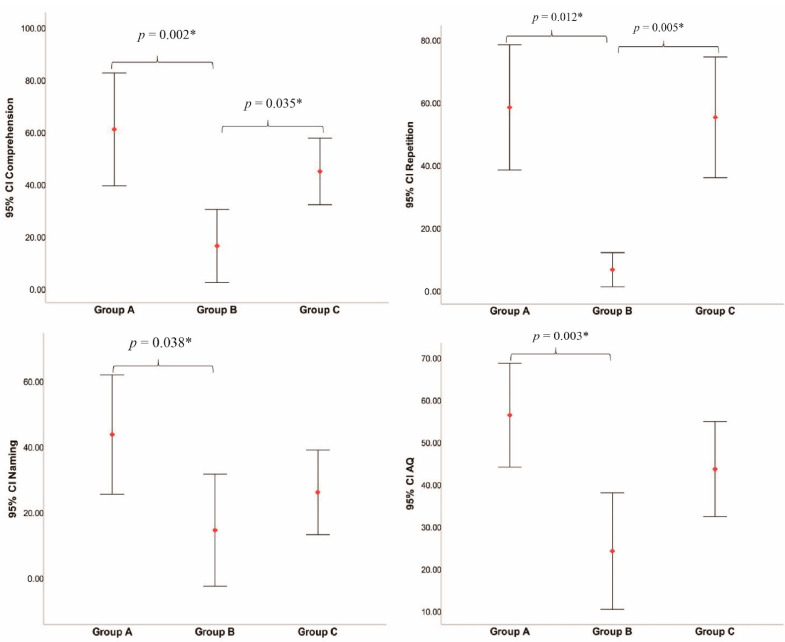
Multiple comparison tests for intergroup differences in language subsets. * Statistical significance level: α = 0.05.

**Table 1 brainsci-12-00907-t001:** Demographic and general clinical characteristics of the patients.

Patient ID	Age/Sex	Education (Years)	Time Post Onset (Days)	Stroke Type	Aphasia Type	Aphasia Severity	Lesion Site	Lesion Volume (cm^3^)	Lesion Load (%)
01	M/52	9	8	Infarction	TSA	mild	Basal ganglia, corona radiate, and centrum semiovale	7.94	0
02	M/49	15	10	Infarction	Broca	moderate	Basal ganglia, frontotemporal parietal lobe, and corona radiata	43.43	3.02
03	F/59	6	12	Infarction	TSA	moderate	Basal ganglia, corona radiate	8.19	0
04	M/36	9	16	Infarction	Broca	moderate	Basal ganglia, temporal lobe, corona radiata, and centrum semiovale	27.79	0
05	M/59	12	17	ICH	Wernicke	severe	Temporal lobe, insula	75.34	0.7
06	F/56	12	18	ICH	Wernicke	severe	Temporal parietal lobe	68.07	26.88
07	F/56	9	81	ICH	Conduction	moderate	Basal ganglia, corona radiata	17.15	0
08	M/38	15	83	ICH	Broca	moderate	Basal ganglia, frontal lobe	27.34	0.02
09	M/55	12	16	Infarction	Global	very severe	Basal ganglia, frontotemporal parietal lobe, and corona radiata	200.50	26.29
10	M/61	12	45	Infarction	Conduction	severe	Frontotemporal parietal lobe	21.02	1.97
11	M/64	12	75	Infarction	Global	very severe	Frontotemporal parietal and occipital lobe	50.87	9.36
12	F/71	6	89	Infarction	Global	very severe	Frontotemporal parietal and occipital lobe	150.50	35.14
13	M/56	15	85	Infarction	Global	severe	Basal ganglia, frontoparietal lobe, and insula	76.59	2.24
14	M/34	12	77	Infarction	Global	very severe	Frontotemporal parietal lobe	54.30	15.19
15	M/61	9	21	Infarction	Anomic	moderate	Basal ganglia, frontal lobe	38.50	2.9
16	M/72	16	23	Infarction	Broca	moderate	Basal ganglia, corona radiate	12.50	0.03
17	F/33	6	5	Infarction	Global	severe	Temporal and parietal lobe, insula	40.26	32.01
18	M/64	9	8	Infarction	Global	severe	Temporal and parietal lobe, insula	11.51	0.89
19	M/69	12	5	Infarction	TSA	moderate	Basal ganglia, temporal and parietal lobe, and corona radiata	14.42	0.28
20	M/48	9	88	ICH	MTA	severe	Basal ganglia, frontotemporal lobe	29.30	0.17
21	F/50	12	15	Infarction	Broca	severe	Temporal and parietal lobe, corona radiata, and insula	10.84	6.33
22	M/43	16	53	Infarction	Anomic	moderate	Frontoparietal lobe, insula	26.95	2.07
23	M/69	6	68	ICH	TMA	severe	Basal ganglia, frontal lobe	46.70	0.19
24	M/64	15	73	Infarction	MTA	severe	Frontoparietal lobe	151.30	21.02
25	M/48	9	24	Infarction	Global	very severe	Frontotemporal parietal lobe, basal ganglia, and insula	1.44	48.30
26	M/50	9	11	Infarction	Wernicke	severe	Temporo-occipital junction, insula	35.26	15.73

Note: F, female; ICH, intracerebral hemorrhage; M, male; MTA, mixed transcortical aphasia; TMA, transcortical motor aphasia; and TSA, transcortical sensory aphasia.

**Table 2 brainsci-12-00907-t002:** The correlation analyses between language performance and MRI measures.

	Lesion Volume		Lesion Load		FA Value in Left AF Segments						Fiber Number in Left AF Segments					
					Anterior		Posterior		Long		Anterior		Posterior		Long	
	ρ	*P*	ρ	*P*	ρ	*P*	ρ	*P*	ρ	*P*	ρ	*P*	ρ	*P*	ρ	*P*
**Spontaneous speech**	−0.447	0.022 *	−0.547	0.004 *	0.348	0.081	0.467	0.016 *	0.525	0.006 *	0.346	0.083	0.281	0.165	0.405	0.040 *
**Comprehension**	−0.393	0.047 *	−0.688	0.000 *	0.237	0.244	0.717	0.000 *	0.521	0.006 *	0.193	0.346	0.687	0.000 *	0.395	0.046 *
**Repetition**	−0.316	0.116	−0.548	0.004 *	0.333	0.097	0.725	0.000 *	0.366	0.066	0.094	0.646	0.629	0.001 *	0.237	0.244
**Naming**	−0.436	0.026 *	−0.642	0.000 *	0.150	0.464	0.587	0.002 *	0.591	0.001 *	0.126	0.541	0.366	0.066	0.381	0.055
**Fluency**	−0.169	0.410	−0.299	0.138	0.363	0.068	0.195	0.340	0.438	0.025 *	0.422	0.032 *	0.055	0.791	0.375	0.059
**AQ**	−0.497	0.010 *	−0.741	0.000 *	0.351	0.079	0.766	0.000 *	0.635	0.000 *	0.240	0.237	0.592	0.001 *	0.445	0.023 *

* Spearman correlation: α = 0.05.

**Table 3 brainsci-12-00907-t003:** The partial correlation analyses between language performance and diffusion indices (controlled variable: lesion load).

	FA Value in Left AF Segments						Fiber Number in Left AF Segments					
	Anterior		Posterior		Long		Anterior		Posterior		Long	
	r	*P*	r	*P*	r	*P*	r	*P*	r	*P*	r	*P*
**Spontaneous speech**	0.161	0.442	0.168	0.421	0.325	0.113	0.307	0.135	0.020	0.926	0.300	0.146
**Comprehension**	0.160	0.445	0.686	0.000 *	0.416	0.039 *	0.243	0.242	0.644	0.001 *	0.333	0.104
**Repetition**	0.198	0.342	0.708	0.000 *	0.270	0.191	0.104	0.620	0.576	0.003 *	0.159	0.447
**Naming**	0.006	0.979	0.488	0.013 *	0.460	0.021 *	0.115	0.583	0.254	0.220	0.295	0.152
**Fluency**	0.231	0.267	0.002	0.993	0.290	0.159	0.362	0.075	−0.107	0.610	0.313	0.127
**AQ**	0.199	0.341	0.695	0.000 *	0.504	0.010 *	0.249	0.231	0.500	0.011 *	0.378	0.062

* Statistical significance level: α = 0.05.

**Table 4 brainsci-12-00907-t004:** The results of the WAB subsets ^#^.

	Group A	Group B	Group C	*p*
**Spontaneous speech**	59.38 ± 12.94	41.67 ± 17.22	45.83 ± 20.43	0.149
**Comprehension**	60.81 ± 25.82	16.17 ± 13.34	44.71 ± 20.05	0.002 *
**Repetition**	58.25 ± 23.89	6.50 ± 5.17	55.08 ± 30.29	0.004 *
**Naming**	43.50 ± 21.73	14.33 ± 16.29	25.92 ± 20.36	0.036 *
**Fluency**	67.50 ± 21.21	50.00 ± 16.73	44.17 ± 23.53	0.076
**AQ**	56.26 ± 14.70	24.07 ± 13.18	43.48 ± 17.72	0.004 *

^#^ Values are given in mean ± SD. α = 0.05. * Statistical significance level: α = 0.05.

**Table 5 brainsci-12-00907-t005:** Multiple comparison tests for the WAB subsets.

	Mean Diff.	95% CI	Adjusted-*p*
**Comprehension**			
Group A–Group B	44.65	15.60, 73.69	0.002 *
Group A–Group C	16.10	−8.45, 40.65	0.311
Group B–Group C	−28.54	−55.43, −1.65	0.035 *
**Repetition**	Test Statistic	Std. Error	Adjusted-*p*
Group A–Group B	11.85	4.13	0.012 *
Group A–Group C	−0.10	3.49	1.000
Group B–Group C	−11.96	3.82	0.005 *
**Naming**			
Group A–Group B	29.17	1.29, 57.04	0.038 *
Group A–Group C	17.58	−5.98, 41.14	0.199
Group B–Group C	−11.58	−37.39, 14.23	0.775
**AQ**			
Group A–Group B	32.20	9.98, 54.41	0.003 *
Group A–Group C	12.79	−5.99, 31.56	0.276
Group B–Group C	−19.41	−39.97, 1.16	0.069

Note: Comprehension: Bonferroni post-hoc test. Repetition: Nemenyi post-hoc test. Naming: Bonferroni post-hoc test. AQ: Bonferroni post-hoc test. * Statistical significance level: α = 0.05.

**Table 6 brainsci-12-00907-t006:** The diffusion indices in the left AF segments ^#^.

	Segments	Group A	Group B	Group C	*p*
**FA value**	Anterior	0.40 ± 0.03	0.00	0.17 ± 0.18	0.001 *
	Posterior	0.39 ± 0.05	0.00	0.25 ± 0.19	0.003 *
	Long	0.42 ± 0.04	0.00	0.13 ± 0.19	0.000 *
**Fiber number**	Anterior	275.25 ± 58.92	0.00	97.58 ± 108.09	0.000 *
	Posterior	320.50 ± 60.88	0.00	181.42 ± 154.14	0.002 *
	Long	309.00 ± 82.90	0.00	64.75 ± 98.12	0.000 *

^#^ Values are given in mean ± SD. * Statistical significance level: α = 0.05.

**Table 7 brainsci-12-00907-t007:** Multiple comparison tests for diffusion indices in the left AF segments.

	Segment	Test Statistic	Std. Error	*p*	Adjusted-*p*
**FA value**	**Anterior**				
	Group A–Group B	14.63	3.92	0.000	0.001 *
	Group A–Group C	9.21	3.31	0.005	0.016 *
	Group B–Group C	−5.42	3.63	0.136	0.407
	**Posterior**				
	Group A–Group B	13.63	4.01	0.001	0.002 *
	Group A–Group C	5.38	3.39	0.112	0.337
	Group B–Group C	−8.25	3.71	0.006	0.019 *
	**Long**				
	Group A–Group B	14.19	3.79	0.000	0.001 *
	Group A–Group C	10.35	3.21	0.001	0.003 *
	Group B–Group C	−3.54	3.51	0.313	0.939
**Fiber number**	**Anterior**				
	Group A–Group B	14.88	3.92	0.000	0.000 *
	Group A–Group C	9.63	3.32	0.004	0.011 *
	Group B–Group C	−5.25	3.63	0.148	0.445
	**Posterior**				
	Group A–Group B	14.38	4.01	0.000	0.001 *
	Group A–Group C	6.63	3.39	0.051	0.152
	Group B–Group C	−7.75	3.72	0.007	0.021 *
	**Long**				
	Group A–Group B	14.75	3.80	0.000	0.000 *
	Group A–Group C	11.58	3.21	0.000	0.001 *
	Group B–Group C	−3.17	3.51	0.368	1.000

* Independent samples Kruskal–Wallis tests with the Nemenyi post-hoc test, statistical significance level: α = 0.05.

## Data Availability

All data used in this study are available from the corresponding author on reasonable request.

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
