# Peer review of "Integrity of the Left Arcuate Fasciculus Segments Significantly Affects Language Performance in Individuals with Acute/Subacute Post-Stroke Aphasia: A Cross-Sectional Diffusion Tensor Imaging Study"

_brainsci, 2022, doi:10.3390/brainsci12070907_

Round 1

Reviewer 1 Report

-        In the abstract the authors state that “Lesion load in language areas is a powerful negative factor”. If this means that left perisylvian lesions result in aphasia and the extent of the lesion is analogous to the severity of aphasia, then I suppose that it’s not something that they could actually report as a novel finding.

-        Mind the wording. See for example line 41, where the authors characterize the contributions of cortical regions as impressive. Also see line 60 (I believe “organization” fits better that “reorganization”).

-        The view that lesion in Broca’s area results in speech production deficits and lesion in Wernicke’s area results in comprehension impairment is oversimplified (see introduction, p2).

-        There have been studies focusing on subcortical structures in aphasia. The claim that they are scarce, could be revised (in the introduction). Moreover, the claim that the importance of white matter tracts in language and aphasia has been highlighted recently does not seem right, given that scholars like Lichtheim, Wernicke, and Geschwind have discussed the crucial role of such fibres in the language network.

-        Description of the dual stream model is a bit outdated. There have been subsequent studies which added more info on language organization in the framework proposed by this model.

-        Characterization of Broca’s and Wernicke’s areas as the “motor center for speech” and “the language comprehension center” respectively is an oversimplification that resembles the obsolete phrenological view, and therefore unacceptable (see line 69).

-        TPO varies significantly across patients. This should be addressed (probably as a limitation of the study)

-        It would be helpful if the authors could provide further information on whether they accounted for lesion area in diffusion images pre-processing via FSL but also AF reconstruction.

-        The authors selected specific “language areas”. However, the selection of these areas is not justified. The authors should fully justify their choice, especially since this choice influences the calculation of the lesion load variable. The justification should be made with caution, since there has been evidence of a widely distributed left-lateralized network supporting language, but also other aspects of cognition.

-        In page 8, the authors state that they run correlation analyses to investigate relationships between variables, among which is sex. However, sex is a binary variable and therefore should not be investigated with correlation analyses.

-        Graphs illustrating the relationships should be presented, to provide a better understanding of the association (whether it is linear, monotonic, etc). Also, graphs illustrating the differences between groups could be helpful (e.g. error-bar charts).

-        In the discussion, the authors extensively discuss the association between lesion load and aphasia severity. They discuss this in relation to the fact that white matter tracts may serve cortico-cortical connections in language-related regions and they highlight the importance of taking into consideration both cortical and subcortical lesions when investigating aphasia. However, given the fact that lesion load is a somewhat of a gross index, the authors cannot actually argue about specific cortico-cortical connections in their sample. Overall, the finding is that more extensive lesions affecting language-related areas will probably result in greater aphasia severity. I do not think that this is a finding that merits publication, especially when it is presented as one of the two main findings of the present study.

-        In the discussion (p13) the authors compare their findings with those of a study with PPA patients. I’m rather sceptical about this comparison, since PPA and post-stroke aphasia are quite different entities, with regard to underlying aetiology and clinical manifestations.

In p. 13, the paper reads: “In the study by Ivanova et al. (2021) [25], microstructural integrity of the left LSAF was associated with auditory comprehension and naming scores, despite not surviving the FDR correction for multiple comparisons. Likewise, in this study, although correlation analysis demonstrated a relationship between the LSAF and language performances including AQ, comprehension, and naming, the specific role of the LSAF could not be further identified in intergroup differences analyses because the diffusion indices on the LSAF in group A was higher than those in group C while the language performances of patients in group A was not different from those in group C”. I do not understand this claim. Following the authors’ rationale, if microstructural integrity of any tract is associated with language performance, then two groups with different structural indices should exhibit different language skills.

Reviewer 2 Report

The authors report a study relating language scores on the WAB to lesion load in anterior and posterior segments of the arcuate fasciculus. They have created lovely images, and the results are sound for the most part.  The findings are not especially novel, but an important confirmation with a few new insights.

I noted several relatively minor weaknesses:

1. Several important articles on the role of the arcuate fasciculus (AF, sometimes referred to as the superior longitudinal fasciculus SLF) and language are not mentioned (e.g. papers by Schlaug et al.) I am aware that not all authors agree the AF and SLF are precisely the same, but most papers refer to the same set of fibers as one or the other, and no studies of which I am aware clearly distinguish them objectively.

2. The Mesulam 2015 paper is an odd choice to cite as evidence that Wernicke’s area is critical for word comprehension, since it claims to show the opposite.  However, it has been heavily criticized.   Recommend selecting a different reference to make your point! There are many good ones, some of which provide direct evidence against the Mesulam 2015 paper.

3.  They state, “Due to its strong relationship with lesion load of cortical language areas, the lesion volume variable was not entered into the partial correlation analyses.” I do not understand the reasoning here (or maybe just the sentence). Please clarify.

4.  “Growing studies” should be changed to “A growing number of studies”

5. Throughout, the word “performances” should be changed to performance.  Performance in this context refers to a single entity (non-countable, like information).  “Performances” is an accurate plural for “performance” only when there are two or more separate things happening (countable events; e.g., there were 2 performances of the play, one in the afternoon and one in the evening.)

6. I am a bit confused about the use of Spearman’s correlations in several cases.  Why use this for dichotomous variables like “type of stroke” (ischemic vs hemorrhagic)?  And for continuous variables, why use Spearman instead of Pearson correlations (was one or more variable not normally distributed)? Should rho be used in place of r?

7. It is necessary to correct for multiple comparisons to draw strong conclusions.

8. In table 2, naming correlation showed p=0.02.  It is missing an *

9.  I do not understand the sentence in line 269, “Considering its strong relationship with AQ, the severity of aphasia was not entered 269 into the correlation analysis.”  Isn’t severity of aphasia equal to AQ (rather than just having a strong relationship with AQ)? How was severity of aphasia measured, if not with AQ?

Reviewer 3 Report

I recommend publication after the authors discuss the broader architectural framing for their findings about the language network; i.e. how does it relate to the finding that posterior temporal regions seem to code for higher-order language structures?

I.e. relating to Matchin & Hickok (2020) model, and Murphy et al. (2022) results? etc...

https://www.jneurosci.org/content/42/15/3216

https://pubmed.ncbi.nlm.nih.gov/31670779/

Reviewer 4 Report

Thanks for recommending me as a reviewer. This paper aimed to investigate the correlation between the left arcuate fasciculus (AF) segments and acute/subacute post-stroke aphasia (PSA). In this paper, when controlled lesion load variable, significant correlations between diffusion indices on the posterior and long segments and comprehension, repetition, naming and aphasia quotient were retained. Multiple comparison tests revealed intergroup differences in diffusion indices on the left AF posterior and long segments as well as these language subsets. No significant correlation was found between the anterior segment and language performances. If authors complete minor revisions, the quality of the study will be further improved.

1. Title: If authors refer to "DTI Study" as "diffusion tensor imaging" in the title, it may help readers to understand.

2. line 69-73: It is recommended to combine this paragraph with the next one. 

3. The introduction section is well written. However, it is too verbose. If the authors describe the introduction section more clearly, it can help readers understand it.

4. line 114-127: Authors should more clearly state the "2.1. Subjects" section. In this section, authors can tabulate the general characteristics of subjects.

5. There are too many footnotes in Table 1, which may confuse the reader.

Round 2

Reviewer 1 Report

The authors have made substantial changes and addressed most of my comments. However, there are some minor changes needed.

-Clarification of point 7 (in the first round of reviews): Apologies for not being clear. TPO means “time post onset”, i.e. the time interval between stroke and assessment. Please address this point. The original point was “TPO varies significantly across patients. This should be addressed.”

-About the graphs (point 11 in the first review): Although I understand the danger of the manuscript being overloaded with too many graphs, I still think that at least some graphs illustrating relationships (as I pointed out in my previous review, whether they are linear, monotonic, etc) and differences (error-bar charts). The authors could choose only their main findings to be illustrated in these graphs.

- Point 13 of the first review: In the discussion (p13) the authors compare their findings with those of a study with PPA patients. I’m rather sceptical about this comparison, since PPA and post-stroke aphasia are quite different entities, with regard to underlying aetiology and clinical manifestations. The authors replied in their response letter: “Thank you for your comments. The underlying aetiology and clinical manifestations of PPA are indeed quite different from post-stroke aphasia. We cited the findings of the study on PPA to provide evidence of the relationship between the AF and language abilities, which may help explain the non-accidentality of our results.”

While I understand their argument, the authors could mention that PPA and post-stroke aphasia are different in terms of underlying aetiology and clinical manifestations.

- P. 5, lines 204-208: the authors claim that they classified their patient’s cohort based on whether AF was fully or partially reconstructed due to lesion load. It seems that other researchers have followed similar approaches in post-stroke aphasia. Perhaps, it would be helpful to mention this in either methods or discussion. See for example:

Kim, S. H., & Jang, S. H. (2013). Prediction of aphasia outcome using diffusion tensor tractography for arcuate fasciculus in stroke. American Journal of Neuroradiology34(4), 785-790.

Kourtidou, E., Kasselimis, D., Angelopoulou, G., Karavasilis, E., Velonakis, G., Kelekis, N., Zalonis, I., Evdokimidis, I., Potagas, C. & Petrides, M. (2021). The role of the right hemisphere white matter tracts in chronic aphasic patients after damage of the language tracts in the left hemisphere. Frontiers in human neuroscience15, 226.

- About point 8 of the first review: I appreciate what the authors say about AF reconstruction in their response letter, however it would be helpful if they could briefly describe the pipeline they followed when accounting for lesion area in pre-processing via FSL, and include this information in the manuscript. 

Reviewer 2 Report

The authors have adequately addressed my concerns

Author Response

Comments and Suggestions for Authors:The authors have adequately addressed my concerns

Response: We are thankful for your comments and suggestions on our manuscript.